# Knowledge Mapping and Research Hotspots of Comorbidities in Psoriasis: A Bibliometric Analysis from 2004 to 2022

**DOI:** 10.3390/medicina59020393

**Published:** 2023-02-17

**Authors:** Shan Huang, Yanping Bai

**Affiliations:** 1Graduate School, Beijing University of Chinese Medicine, Beijing 100105, China; 2Department of Dermatology, China-Japan Friendship Hospital, Beijing 100029, China

**Keywords:** psoriasis, comorbidity, research hotspots, bibliometrics, visualization analysis

## Abstract

*Background and Objectives*: Psoriasis is a chronic inflammatory disease whose impact on health is not only limited to the skin, but is also associated with multiple comorbidities. Early screening for comorbidities along with appropriate treatment plans can provide a positive prognosis for patients. This study aimed to summarize the knowledge structure in the field of psoriasis comorbidities and further explore its research hotspots and trends through bibliometrics. *Materials and Methods*: A search was conducted in the core collection of the Web of Science for literature on comorbidities of psoriasis from 2004 to 2022. VOSviewer and CiteSpace software were used for collaborative network analysis, co-citation analysis of references, and keyword co-occurrence analysis on these publications. *Results*: A total of 1803 papers written by 6741 authors from 81 countries was included. The publications have shown a progressive increase since 2004. The United States and Europe were at the forefront of this field. The most prolific institution was the University of California, and the most productive author was A. Armstrong. Research has focused on “psoriatic arthritis”, “metabolic syndrome”, “cardiovascular disease”, “psychosomatic disease”, “inflammatory bowel disease”, “prevalence”, “quality of life”, and “risk factor” in the past 18 years. Keywords such as “biologics” and “systemic inflammation”, have been widely used recently, suggesting current research hotspots and trends. *Conclusions*: Over the past 18 years, tremendous progress has been made in research on psoriasis comorbidity. However, collaborations among countries, institutions, and investigators are inadequate, and the study of the mechanisms of interaction between psoriasis and comorbidities and management of comorbidities is insufficient. The treatment of comorbidities with biologic agents, screening of comorbidities, and multidisciplinary co-management are predicted to be the focus of future research.

## 1. Introduction

Psoriasis, a chronic, inflammatory, relapsing skin disease mediated by T cells and characterized by erythematous scaling, affects more than 125 million individuals worldwide and imposes a huge burden on patients [1,2]. With overlapping genetic loci, abnormal immunomodulatory mechanisms and a cross-cutting chronic inflammatory process, the health effects of psoriasis are not confined to the skin [3]. As research progresses, concepts, such as “psoriasis march” [4] and “systemic psoriasis” [5], have been proposed, revealing that psoriasis is a systemic disease that may be associated with several comorbidities, including psoriatic arthritis, metabolic syndrome, cardiovascular disease, inflammatory bowel disease, mental psychological diseases, malignant tumors, infections, and several skin diseases [3,6]. Approximately 57.9% of patients with psoriasis have at least one comorbidity [7], which not only influences their treatment options but also decreases their quality of life and shortens their life expectancy [8,9]. Therefore, a comprehensive and systematic review of research results in the field of psoriasis comorbidities and a summary of the development trends and research hotspots in this field are essential.

Bibliometrics is the discipline of describing, evaluating, and predicting the current status and trends of a field using mathematical and applied statistical methods related to the number of publications in the literature [10]. Compared to traditional reviews and meta-analyses that focus on a particular aspect, bibliometrics tend to provide a comprehensive and visualized demonstration of the knowledge structure in the field, such as overall trends in research, core authors and institutions, evolution of the subject, and current hot frontiers. However, to the best of our knowledge, no bibliometric study on psoriasis co-morbidities has been conducted yet. To fill this gap, this study used VOSviewer and CiteSpace to visualize the publications related to psoriasis comorbidities in the core collection of the Web of Science (WOSCC). Co-presence analysis, co-citation analysis, cluster analysis, and burst detection were used to summarize the research hotspots and trends in this field and draw a knowledge map. It is intended to provide accurate and comprehensive information in this area for clinicians and researchers.

## 2. Materials and Methods

### 2.1. Data Sources and Search Strategies

We conducted a literature search on the WOSCC database on 30 September 2022, and this process was completed independently by two researchers to ensure the accuracy of the results. Specific search strategy: TS = (psoriasis) AND TS = (multimorbid* OR comorbid* OR polymorbid* OR multi-morbid* OR co-morbid* OR poly-morbid*) [11] AND PY = (2004 to 2022). The language of the article was limited to English, and the type of article was limited to article or review. A total of 2413 documents was initially screened. Thereafter, two researchers performed a secondary screening of the obtained articles individually by reading the paper titles and abstracts to ensure the relevance of the literature to the research topic. Disagreements were resolved by discussion between the two researchers. The “Full Record and Cited References” of the filtered publications were exported in “plain text file format” and imported into CiteSpace to remove duplicate data, resulting in 1803 documents (Figure 1).

### 2.2. Data Analysis and Visualization

In this study, CiteSpace (6.1.R3), and VOSviewer (V. 1.6.18) were used for data analysis and visualization. VOSviewer [12] is a visual analysis software that enables the analysis and visualization of cooperative, co-occurrence and co-citation networks of research data. In this study, VOSviewer was used to build the collaboration network of countries, institutions, and authors, and further visualize the results through Scimago Graphica. CiteSpace [13] is a bibliometric software that is used not only to analyze collaboration, co-occurrence, and co-citated networks, but also to investigate the evolution of domain knowledge structures and frontier dynamics. In this study, CiteSpace was used to perform cooperation network analysis, reference co-citation analysis, keyword co-occurrence analysis, and to detect the keywords with the strongest bursts. The specific parameter settings are shown in the Appendix A.

## 3. Results

### 3.1. Global Trends in Annual Publications

The number of annual publications reflects the trends in the field. Between 2004 and 2022, a cumulative total of 1803 publications in the field of psoriasis comorbidities was included in the WOSCC, containing 1348 papers and 455 reviews. An overall upward trend in the number of publications was observed (Figure 2). According to the WOS citation report, these texts have been cited 5240 times cumulatively, and the number of citations increased yearly (Figure 2), suggesting that the field of psoriasis comorbidities is flourishing, with increasing scholarly attention.

### 3.2. Cooperation Network

#### 3.2.1. Country–Region Cooperation Network

Data on countries and regions of publications were analyzed using VOSviewer and CiteSpace and visualized using Scimago Graphica for countries and regions with a total number of ≥20 publications (Figure 3A). The results showed that 81 countries and regions conducted research on psoriasis comorbidities, among which the United States had the most publications (499), followed by Italy (269) and Germany (187). Most of the high-producing countries were concentrated in the European and North American regions. Betweenness centrality (BC) [14] is an indicator of the strength of linkages, with BC ≥0.1 suggesting close links between a node and other nodes. Although it was the most prolific country, the United States (BC = 0.06) had no strong collaborations with other countries. The United Kingdom (BC = 0.25), the fourth most prolific publisher, has developed the strongest international collaboration and has established collaborations with highly productive countries, such as the United States, Italy, Germany, and Australia.

#### 3.2.2. Institutional Cooperation Network

Data on institutions in the articles were analyzed using VOSviewer and CiteSpace, and those with ≥5 publications were visualized (Figure 3B). The results showed that 2669 institutions were involved in research on psoriasis comorbidities, and high-production institutions were dominated by universities. The University of California in the United States had the highest number of publications (92), followed by Harvard University (66) and the University of Copenhagen in Denmark (61). The density of co-occurrence profiles reflects the cooperation among nodes in general. The co-occurrence mapping density in this study was 0.025, suggesting a low level of overall collaboration among institutions. However, closer collaborations were formed in the top few institutions for article volume, such as the University of California (BC = 0.10), the University of Toronto (BC = 0.10), and National Taiwan University Hospital (BC = 0.19).

#### 3.2.3. Author Collaboration Network

VOSviewer and CiteSpace were used to analyze author information in the articles and visualize the top 213 core authors with ≥5 publications (Figure 3C). The results showed that 6741 academics had published in the field of psoriasis co-morbidities. Among them, A. Armstrong was the most prolific author, with 36 publications since 2012 and a high betweenness centrality (BC = 0.13), making him the most influential scholar in this field. C. Griffiths (BC = 0.02) has been investigating psoriasis comorbidities since the starting year of the search, 2004, and has accumulated the second highest number of 35 publications. Additionally, P. Gisondi (BC = 0.11) has accumulated 35 publications, while collaborating more closely with other authors. However, in the overall collaborative network, the plot density was only 0.0233, suggesting a lack of collaboration among most authors.

### 3.3. References Co-Cited

References are the knowledge base of research, and reference co-citation analysis can reflect the knowledge structure of the field, trace the research evolution of the discipline, and indicate current research hotspots. The references of the obtained publications were visually analyzed using CiteSpace, and the key references with ≥10 citations were visualized (Figure 4A). The results showed that 45,156 references were cited in 1803 psoriasis co-morbidity publications. The most cited references were “Psoriasis and comorbid diseases: Epidemiology” [15] written by Takeshita J, and “Global epidemiology of psoriasis: a systematic review of incidence and prevalence” [16] written by the Identification and Management of Psoriasis and Associated Comorbidity (IMPACT) project team. Both were studies on the epidemiology of psoriasis that had been cited 102 times. Notably, references with high intermediary centrality, such as “Psoriasis and comorbid diseases: Epidemiology” [15] and “Psoriasis and major adverse cardiovascular events: a systematic review and meta-analysis of observational studies” [17], are circled in purple in Figure 4A. These references may be located among different clusters, indicating that they may be landmark studies in the field.

These references were further clustered (Figure 4B), and a timeline graph was drawn to demonstrate the research process in the discipline (Figure 4C). The quality of clustering was evaluated using the Modularity Q and Mean Silhouette. In our study, the Modularity Q was 0.7185 and the Mean Silhouette was 0.894, indicating that the cluster structure was clear and the clustering result was reasonable and reliable. Based on the clustering results, the references were mainly clustered into 11 clusters. The timeline diagram further demonstrates the changing of the knowledge structure in the field. Researchers had an early focus on “metabolic syndrome” (#2), “coronary revascularization” (#3), “psychological” (#7), and “health insurance” (#8), followed by “psoriatic arthritis” (#0), “autoimmune disease” (#6), “arthritis” (#9), and “adiponectin” (#10). “Adiponectin” (#10) was understood as a mechanistic exploration of “metabolic syndrome” (#2). The recent focuses, including “cardiovascular risk” (#1), were further extensions and expansions of “coronary revascularization” (#3); and “psychiatric disorders” (#4) is an extension of “psychological” (#7), whereas “older adults” (#5) is a focus of research emerging recently.

### 3.4. Keyword Co-Occurrence

Keywords are a distillation of the literature, and keyword analysis can show the focus of research and cutting-edge trends in the field. The keywords, such as “psoriasis”, “plaque psoriasis”, “co-morbidities”, and “comorbidities”, that affected the analysis were removed. Thereafter, CiteSpace was used to analyze the keywords for co-occurrence and visualize the keywords with ≥10 occurrences (Figure 5A). The top 10 keywords with the highest number of occurrences were “prevalence”, “quality of life”, “metabolic syndrome”, “rheumatoid arthritis”, “psoriatic arthritis”, “cardiovascular disease”, “arthritis”, “risk factor”, “epidemiology”, and “therapy”, which revealed the focus of research in this field. Keywords with high intermediary centrality were “atopic dermatitis”, “cardiovascular disease”, “clinical feature”, and “TNF-alpha”, suggesting possible turning points in research. Further clustering of these keywords was performed, and a timeline plot was drawn (Figure 5B). The results showed that these keywords were clustered into five categories: metabolic syndrome (#0), quality of life (#1), psoriatic arthritis (#2), epidemiology (#3), and inflammatory bowel disease (#4). Modularity Q > 0.5 and Mean Silhouette > 0.3 indicated a reasonable and reliable clustering structure. The timeline diagram illustrates the evolution of these five themes, and the results show that several important keywords emerged in the early years of these five types of research clusters, most of which have been used till today, and new keywords have emerged subsequently. However, the scale of research around new keywords is much less than that of the earlier emergence.

The burst detection of the above keywords was performed using CiteSpace to capture the rapid increase of keywords over a period of time and reveal research hotspots in the field of psoriasis comorbidities. The burst keywords were presented in a chronological order (Figure 5C). Notably, the keywords, “coronary heart disease”, “metabolic syndrome”, “infliximab”, “atherosclerosis”, “Crohn’s disease”, and “systemic inflammation”, had longer outbreaks, indicating a focus of research on the area of psoriasis comorbidities. The high burst of keywords, “myocardial infarction”, “vascular disease”, “coronary artery disease”, and “TNF-α”, suggest that these areas have undergone extensive research and received a high level of attention over a specific time period. Additionally, “systemic inflammation”, “vascular inflammation”, “secukizumab”, “axial arthritis”, “stress”, “biologics”, and “cutaneous malignant melanoma” still have high outbreak values to date, suggesting a current research hotspot or cutting-edge trend in this field.

## 4. Discussion

In 1897, Professor Struss first pointed out the correlation between psoriasis and diabetes, initiating research in the field of psoriasis comorbidities [18]. Currently, psoriasis is considered to be a systemic disease, and chronic inflammation underlies the pathology of psoriasis and its comorbidities [19]. Psoriasis comorbidities, which lead to higher cost of living, poorer quality of life, and worse prognosis, have attracted much attention from scholars, and several relevant papers have been published [8]. In the face of explosive growth of publications, it is difficult and important to have a comprehensive and systematic understanding and to maintain sensitivity to research hotspots in the field of psoriasis comorbidities. Instead of traditional reviews and meta-analyses, bibliometrics has the advantage of showing trends in the research field and analyzing research hotspots [10]. This study is the first bibliometric study in the field of psoriasis comorbidities, which demonstrates the knowledge structure and its evolution over the past 18 years, summarizes research hotspots, and predicts future research trends of the field.

The number of annual publications reflects the trends in a discipline, and from this perspective, psoriasis comorbidities have developed rapidly in the last decade. Psoriasis has transformed from a cutaneous disease to a systemic disease state characterized by systemic inflammation, and an increasing number of diseases have been included in the category of psoriasis comorbidities [20]. However, high-impact articles in this field are dominated by reviews, suggesting a lack of high-quality original research. Therefore, more evidence-based prospective studies and mechanism exploration studies are needed to advance this field in the future.

The collaborative network of countries and regions, institutions, and authors provides a picture of the leading researchers in this field at different levels. The United States, Italy, and Germany are leaders in this field. Moreover, most of the highly productive institutions and authors are concentrated in these regions, suggesting more medical and research resources and a greater awareness of health management. Additionally, according to previous research, these regions have a higher prevalence of psoriasis, which explains the differences in geographical contribution to some extent [21]. However, collaboration is insufficient at either the national and regional, institutional, or author level. Although a few fixed collaborative networks have been formed, cooperation among most research groups is scarce; especially international cooperation among institutions, which remains to be strengthened.

In this study, the co-citation analysis of references provides an overview of the foundation and development of research in the field of psoriasis comorbidities (Figure 4). To better summarize the research in this area, we reviewed the key literature in each cluster.

The relationship between psoriasis and metabolic disorders (#2) is complex. Several metabolic disorders can increase the risk of psoriasis and affect its treatment. For example, obesity is considered to be one of the risk factors for psoriasis, and the frequency of psoriasis and PASI scores tend to increase with increasing BMI [22,23]. Weight reduction through diet control, physical exercise, or bariatric surgery has a positive effect on psoriasis in overweight patients [24,25,26]. Additionally, obesity has the potential to reduce the efficacy of TNF-α inhibitors and increase the risk of liver fibrosis associated with methotrexate [27,28]. Similarly, patients with diabetes have a higher prevalence of psoriasis, which may be associated with insulin use, and DPP-4 inhibitor, a drug used for diabetes, has been shown to ameliorate psoriasis [29,30]. Moreover, several studies have confirmed that patients with psoriasis, especially severe psoriasis, suffer from an increased risk of metabolic diseases, such as obesity, diabetes and its complications, hypertension, dyslipidemia, metabolic syndrome, and gout [31,32,33]. Improvement in psoriasis is usually accompanied by the remission of insulin resistance and a decrease in leptin levels [34]. TNF-α inhibitors have been shown to reduce the prevalence of diabetes and metabolic syndrome in patients with psoriasis through clinical studies, but are associated with an increased risk of weight gain, the paradox of which requires further investigation [35,36].

The association between psoriasis and cardiovascular disease (#3, #1) has been demonstrated in several studies. Gelfand et al. identified psoriasis as an independent risk factor for myocardial infarction through a prospective cohort study that found a significantly increased risk of myocardial infarction in patients with psoriasis, especially in young patients with severe psoriasis [37]. Subsequently, the association between psoriasis and atrial fibrillation and ischemic stroke was confirmed [38,39]. Boehncke et al. proposed a “psoriasis progression” model to explain the mechanisms underlying the complications of psoriasis with cardiovascular disease [4]. The presence of chronic inflammation in patients with psoriasis induces an imbalance of adipokines, such as leptin, which in turn leads to insulin resistance and affects the function of the vascular endothelium, leading to metabolic syndrome and atherosclerosis, and eventually myocardial infarction or stroke. Cardiovascular disease is the primary cause of death in patients with psoriasis, and the life expectancy of patients with moderate-to-severe psoriasis has been demonstrated to be reduced by approximately 5 years due to cardiovascular disease [40]. Recent studies have found that systemic therapy can reduce the risk of cardiovascular disease and improve or even reverse existing disease to some extent in patients with psoriasis, and this efficacy correlates with the cumulative exposure to therapy [41,42]. However, some studies have a contrasting view, and the heterogeneity of this requires further confirmation in large prospective studies [43].

Patients with psoriasis often suffer from comorbid psychiatric disorders (#7, #4). Hedemann et al. found that patients with psoriasis were 1.5 times more likely to experience depression than patients without psoriasis. Additionally, anxiety symptoms (20–50%), schizophrenia (2.82%), and suicidal ideation (12.7%) were more prevalent in patients with psoriasis [44]. The association between psoriasis and alexithymia, posttraumatic stress syndrome, personality disorders, addictive behaviors, obsessive–compulsive disorders, and restless legs syndrome has been demonstrated [45,46,47,48]. Further studies found that patients with psoriasis who were female, young, had a young age of onset, and had skin lesions in sensitive or visible areas were at increased risk of psychosocial disorders [49]. While scholars previously attributed the psychological comorbidities of psoriasis to the psychological stigma and reduced quality of life caused by skin damage, subsequent studies have found overlapping genetic loci and cross-cutting inflammatory response pathways between the two [50,51]. Additionally, biologics have been shown to reduce the risk of depression in patients with psoriasis, possibly due to a reduction in systemic inflammation [52].

Medicare databases (#8) are one of the important data sources for psoriasis comorbidity research and the main research vehicle for economic expenditure. For example, Feldman et al. retrospectively investigated comorbidities and healthcare expenditures among US patients with psoriasis through a large claims database, and the results showed that hypertension (34.3%), hyperlipidemia (33.5%), and cardiovascular disease (17.7%) were the most common comorbidities, with comorbidities resulting in more frequent medical visits and a higher healthcare burden among patients with psoriasis [53].

The prevalence of psoriatic arthritis (#0, #9), the leading cause of teratogenicity in psoriasis, varies from 14% to 22.7% in patients with psoriasis [54]. Compared to patients with psoriasis, patients with psoriatic arthritis generally have a higher incidence of comorbidities and a poorer quality of life [55]. Early diagnosis of psoriatic arthritis has received much attention from scholars. Nail lesions have been reported to be an early predictive sign of psoriatic arthritis [56], and detection of anti-LL37 antibodies can contribute to the identification of PsA to a certain extent [57]. In addition to psoriatic arthritis, other joint diseases, such as ankylosing spondylitis and rheumatoid arthritis, have been found to be associated with psoriasis, and abnormal activation of adaptive immunity is their common feature [58]. Statistically, only 41% of clinical joint symptoms are caused exclusively by psoriatic arthritis [59]. Due to the similarity of clinical manifestations, distinguishing the joint manifestations of patients with psoriasis presents a challenge to clinicians.

Patients with psoriasis have been reported to suffer from a higher frequency of autoimmune diseases (#6), such as uveitis [60], inflammatory bowel disease [61], multiple sclerosis [62], Hashimoto’s thyroiditis [63], bullous pemphigoid [64], vitiligo [65], and alopecia areata [66]. The association between them is exerted through intricate immunopathological pathways and mechanisms, which have not yet been fully elucidated [67]. Biologically targeted therapies have been proven not only for the treatment of psoriasis but also for other autoimmune comorbidities. For example, TNF-α inhibitors have been approved for the treatment of inflammatory bowel disease, rheumatoid arthritis, and ankylosing spondylitis [35]. However, IL-17 inhibitors have the risk of inducing and aggravating IBD, and caution should be exercised in clinical use [35].

Psoriasis is usually lifelong, and elderly patients (#5) have a higher prevalence of psoriasis co-morbidities due to their specific physiopathological characteristics. The prevalence of co-morbidity in patients aged 41–60 years with psoriasis was 64.9%, and this prevalence increased to 89.8% in patients aged 61–80 years [7]. Therefore, extra attention should be paid to screening for comorbidities among elderly patients, and medications that have a risk of aggravating comorbidities, such as cyclosporine, which may induce and aggravate hypertension, and avastin, which may cause dyslipidemia and hypertension, should be used with caution. Biological agents are currently recommended as safer and more effective treatments, but appropriate screening should be performed prior to treatment [68].

Keyword analysis presents an overview of research and cutting-edge hotspots in the field of psoriasis comorbidities. In addition to previously recognized comorbidities, emerging comorbidities, such as osteoporosis and fractures, chronic obstructive pulmonary disease, periodontitis, atopic dermatitis, maternal adverse events, male erectile dysfunction, obstructive sleep apnea, restless legs syndrome, skin cancers, vitiligo, baldness, and COVID-19, have been shown to be associated with psoriasis [3,69]. Notably, “atopic dermatitis”, “cardiovascular disease”, “clinical features”, and “TNF-α” were found to have a high betweenness centrality in the keyword co-occurrence mapping, which may represent intersections between disciplines and promote the generation of new research directions. 

Atopic dermatitis was previously thought to involve a mechanism of inflammatory response centered on Th2/Th22 cells, unlike psoriasis, which is characterized by Th1/Th17-type inflammatory response [70]. Subsequent studies have found that psoriasis and atopic dermatitis can be classified as two ends of the same spectrum, referred to as “psoriatic dermatitis” [71], and that the two are interchangeable, especially in patients applying biologic agents, which have been attributed to immune drift [72,73]. The association between cardiovascular disease and psoriasis is not only exerted through cross-cutting inflammatory responses, but additionally, comorbidities of psoriasis, such as hypertension, diabetes, obesity, and dyslipidemia, are responsible for cardiovascular disease, rendering it an interdisciplinary concern. Studies on clinical features of diseases provide a basis for identifying comorbidities. For instance, patients with early age of onset, severe lesions, and a family history of cardiovascular disease were found to be more likely to develop adverse cardiovascular events [40], and patients with nail involvement and perianal rashes were more likely to develop psoriatic arthritis [74]. Ancillary examinations and laboratory tests, such as fluorodeoxyglucose F-18 positron emission tomography/computed tomography, is used to measure aortic vascular inflammation, which has been adopted as an indicator of cardiovascular risk and vascular disease [75]. Additionally, adipokines and leptin, reflecting the overall level of inflammation, can be used as screening and predictive indicators for psoriasis comorbidities [76,77]. Another research hotspot, TNF-α, is one of the key pathogenic factors of psoriasis. Upon external stimulation, dendritic cells in susceptible individuals are activated to secrete inflammatory mediators, such as TNF-α, and in turn, chemotactic downstream T lymphocytes induce a local inflammatory response environment, stimulate abnormal proliferation of keratin-forming cells, and promote dermal capillary proliferation, constituting the pathological basis for the development of psoriasis [2]. Additionally, TNF-α is involved in the development of various comorbidities, such as coronary atherosclerosis, obesity, ankylosing spondylitis, rheumatoid arthritis, inflammatory bowel disease, uveitis, and septic sweat gland [78,79,80].

The keyword burst predicts possible future frontier hotspots from another perspective. Psoriasis is a systemic inflammatory disease, and several studies have explored the link between psoriasis and comorbidities from different inflammatory mediators and signaling pathways. However, overall, the current research is too fragmented and not well connected. Therefore, the systemic inflammatory mechanism of psoriasis and its connection with comorbidities is yet to be elucidated; more in-depth research is needed in this field in the future. Biologics represent a landmark leap in the history of psoriasis treatment; in addition to significantly improving the skin damage of psoriasis, they have shown benefits for comorbidities in metabolic diseases, cardiovascular diseases, autoimmune diseases, and psychiatric disorders [81,82]. However, some conflicting findings mean that future higher quality studies are needed to evaluate the efficacy of biologics for comorbidities.

The treatment and management of psoriasis as a systemic disease is known to be important and complex. Currently, the treatment of psoriasis is not only focused on skin damage but also on the improvement of comorbidities and quality of life [83]. Although the concept of multidisciplinary co-management has been proposed for a while, this multidisciplinary management pattern has not been widely implemented in clinical practice, and the screening and management of psoriasis comorbidities need to be improved [84,85,86].

## 5. Conclusions

This is the first bibliometric study in the field of psoriatic comorbidities, providing a comprehensive overview of the knowledge structure, historical course, and cutting-edge trends in the field. Research on psoriasis comorbidities has received much attention from scholars and has been developing rapidly. Cardiovascular diseases, metabolic diseases, arthritis, and psychological disorders have been the main areas of focus of research to date, while other emerging comorbidities, treatment with biological agents, screening for co-morbidities, and integrated multidisciplinary management have received increasing attention and are possible trends in the future.

## Figures and Tables

**Figure 1 medicina-59-00393-f001:**
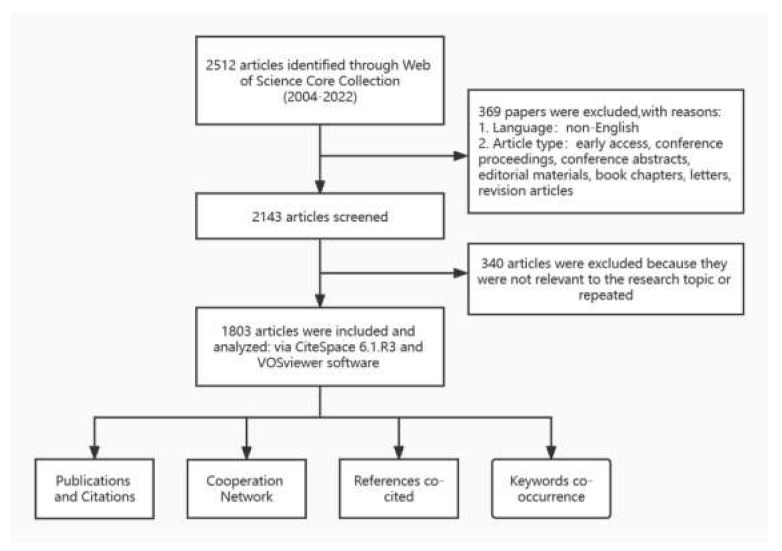
Literature search, screening, and analysis process.

**Figure 2 medicina-59-00393-f002:**
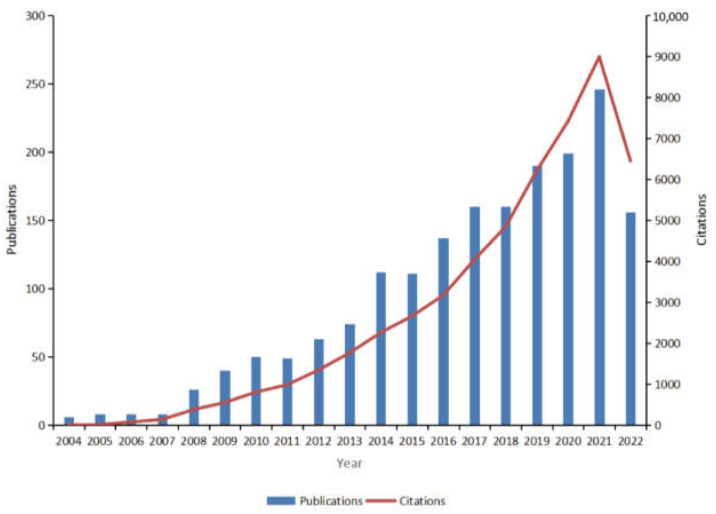
Global trends in publications and total citations for research on comorbidities in psoriasis (2004–2022).

**Figure 3 medicina-59-00393-f003:**
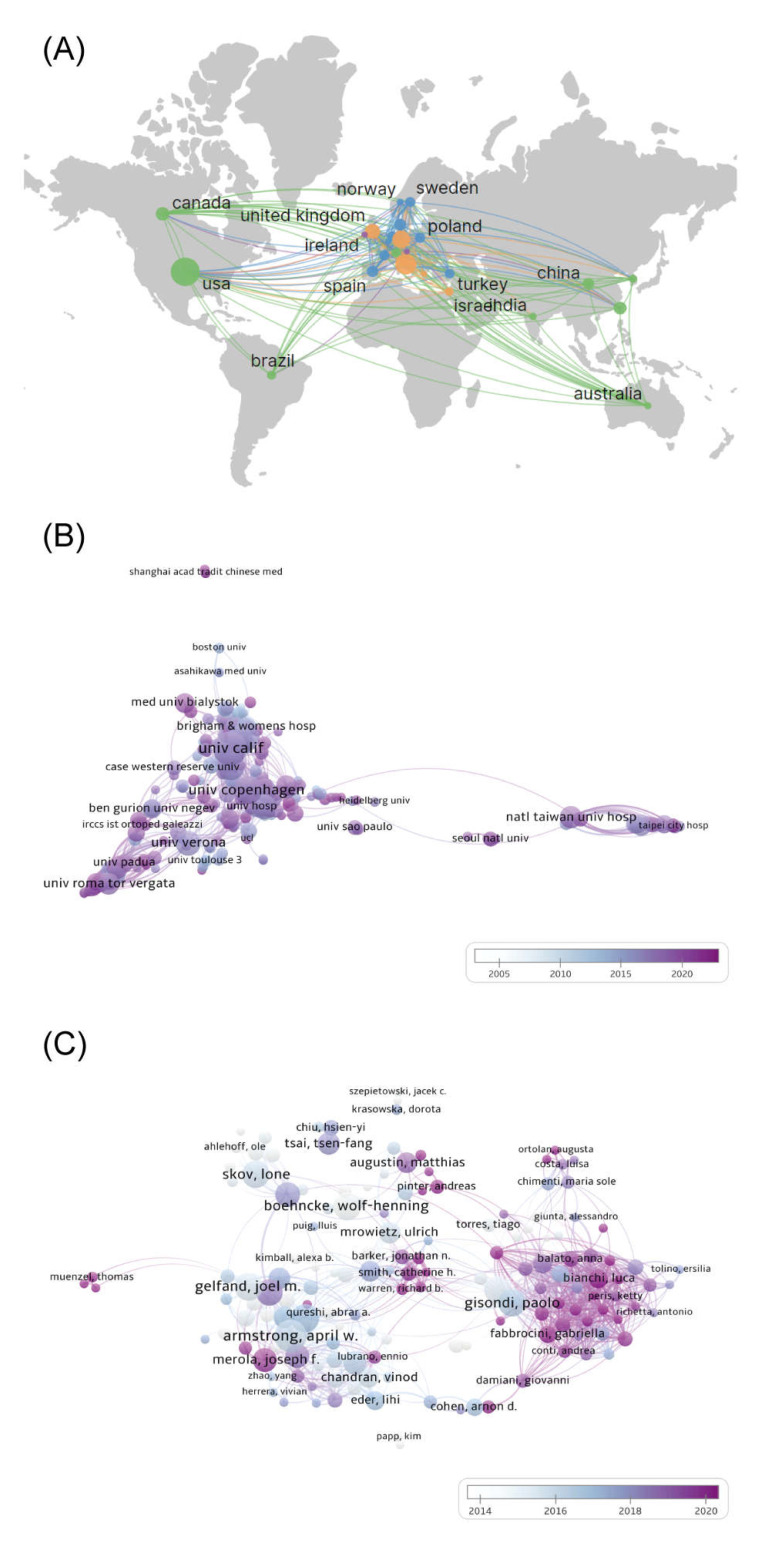
Collaboration network in psoriasis co-morbidities. (**A**) Country and region (**B**) Institutional (**C**) Author collaboration network.

**Figure 4 medicina-59-00393-f004:**
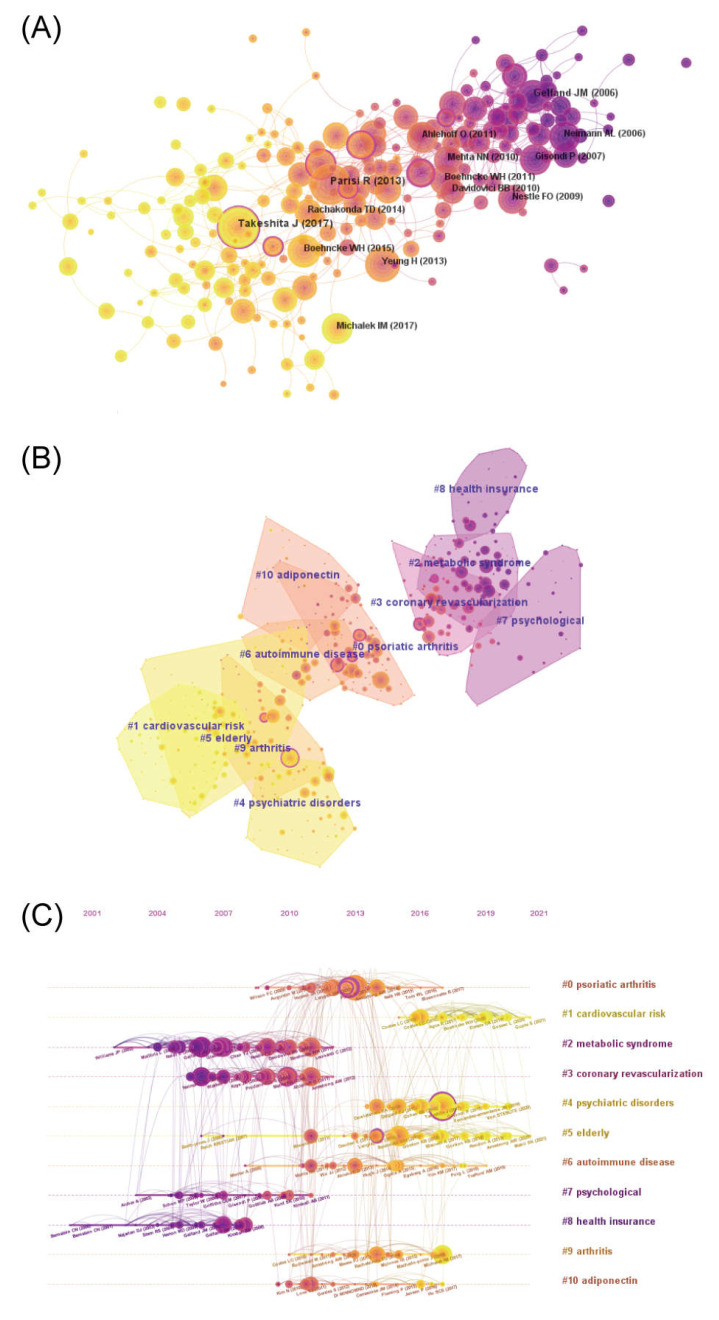
References co-cited in psoriasis co-morbidities. (**A**) The co-cited network of references (**B**) Cluster of reference co-cited (**C**) Timeline of references co-cited.

**Figure 5 medicina-59-00393-f005:**
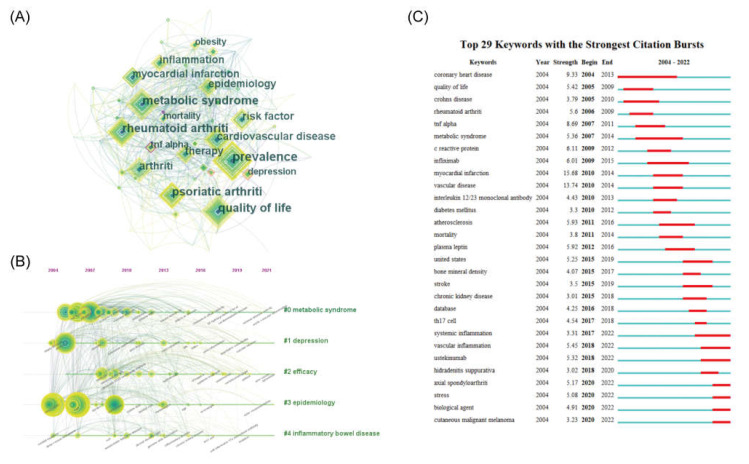
Keywords in psoriasis co-morbidities. (**A**) The co-occurrence network of keywords (**B**) Timeline for clustering of keywords (**C**) The top 29 most bursting keywords.

## Data Availability

Not applicable.

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
