# Peer review of "Knowledge Mapping and Research Hotspots of Comorbidities in Psoriasis: A Bibliometric Analysis from 2004 to 2022"

_medicina, 2023, doi:10.3390/medicina59020393_

Round 1
Reviewer 1 Report
Authors aimed to summarize the knowledge structure in the field of psoriasis comorbidities and further explore its research hotspots and trends through bibliometrics. Materials and Methods: A search was conducted in the core collection of Web of Science for literature on comorbidities of psoriasis from 2004 to 2022. VOSviewer and CiteSpace software were used for collaborative network analysis, co-citation analysis of references, and keyword co-occurrence analysis on these publications. Results: A total of 1,803 papers written by 6,741 authors from 81 countries were included. The publications have shown a progressive increase since 2004. The United States and Europe were at the forefront of this field. The most prolific institution was University of California, and the most productive author was A. Armstrong. Research has focused on “psoriatic arthritis”, “metabolic syndrome”, “cardiovascular disease”, “psychosomatic disease”, “inflammatory bowel disease”, “quality of life”, and “risk factor” in the past 18 years. Keywords, such as “biologics” and systemic inflammation”, have been widely used recently, suggesting current research hotspots and trends. Conclusions: Over the past 18 years, tremendous progress has been made in research on psoriasis comorbidity research. However, collaborations among countries, institutions, and investigators are inadequate, and the study of the mechanisms of interaction between psoriasis and comorbidities, and management of comorbidities is insufficient. The treatment of comorbidities with biologic agents, screening of comorbidities, and multidisciplinary co-management are predicted to be the focus of future research.
Author Response
We are so grateful for your review. Psoriasis is a systemic disease that not only affects patients' quality of life, but also shortens their life expectancy. We provide a comprehensive and systematic review and analysis of research achievements in the field of psoriasis co-morbidities and summarize the trends and research hotspots in this field through bibliometrics. The results of the study show that: Research on psoriasis comorbidities has received much attention from scholars and has been developing rapidly. Cardiovascular diseases, metabolic diseases, arthritis, and psychological disorders have been the main focus of research to date, while other emerging comorbidities, treatment with biological agents, screening for co-morbidities, and integrated multidisciplinary management have received increasing attention and are possible trends in the future.
Thank you for your attention and time. Looking forward to receiving your response.
Reviewer 2 Report
An interesting bibliometric study evaluating the various psoriatic comorbidities; I enjoyed reading this article; only minor queries before publication:
line 38.-43, some other reference is required, such as: doi: 10.1371/journal.pone.0241575
Thank You
Author Response
We are very grateful for your kind advice. Psoriasis is a systemic disease associated with multiple co-morbidities, and we have added relevant references (doi:10.1371/journal.pone.0241575) in lines 38-43 to support this contention.
Thank you for your attention and time. Looking forward to receiving your response.